# BNT162b2 Booster Dose Elicits a Robust Antibody Response in Subjects with Abdominal Obesity and Previous SARS-CoV-2 Infection

**DOI:** 10.3390/vaccines11121796

**Published:** 2023-11-30

**Authors:** Alexis Elias Malavazos, Carola Dubini, Valentina Milani, Sara Boveri, Chiara Meregalli, Caterina Bertolini, Carola Buscemi, Rosanna Cardani, Laura Valentina Renna, Manuel Bruno Trevisan, Valentina Scravaglieri, Maria Teresa Cuppone, Lorenzo Menicanti, Elena Costa, Federico Ambrogi, Chiara Ruocco, Michele Carruba, Gianluca Iacobellis, Enzo Nisoli, Massimiliano Marco Corsi Romanelli

**Affiliations:** 1Endocrinology Unit, Clinical Nutrition and Cardiometabolic Prevention Service, IRCCS Policlinico San Donato, San Donato Milanese, 20097 Milan, Italychiara.meregalli@grupposandonato.it (C.M.);; 2Department of Biomedical, Surgical and Dental Sciences, University of Milano, 20122 Milan, Italy; 3Laboratory of Biostatistics and Data Management, Scientific Directorate, IRCCS Policlinico San Donato, San Donato Milanese, 20097 Milan, Italy; 4Vaccination Unit, ASST Fatebenefratelli Sacco, 20157 Milan, Italy; 5Unit of Internal Medicine, V. Cervello Hospital, 90146 Palermo, Italy; 6Clinical Nutrition Unit, Department of Health Promotion, Maternal and Childhood, Internal and Specialized Medicine of Excellence (PROMISE), University of Palermo, 90100 Palermo, Italy; 7Biobank BioCor, IRCCS Policlinico San Donato, San Donato Milanese, 20097 Milan, Italylauravalentina.renna@grupposandonato.it (L.V.R.); 8Scientific Directorate, IRCCS Policlinico San Donato, San Donato Milanese, 20097 Milan, Italy; 9Service of Laboratory Medicine, IRCCS Policlinico San Donato, San Donato Milanese, 20097 Milan, Italy; 10Department of Clinical Sciences and Community Health, University of Milano, 20122 Milan, Italy; 11Centre for Study and Research on Obesity, Department of Medical Biotechnology and Translational Medicine, University of Milan, 20129 Milan, Italy; chiara.ruocco@unimi.it (C.R.); enzo.nisoli@unimi.it (E.N.); 12Division of Endocrinology, Diabetes and Metabolism, Department of Medicine, University of Miami, Miami, FL 33136, USA; giacobellis@med.miami.edu; 13Department of Biomedical Sciences for Health, University of Milan, 00133 Milan, Italy; mmcorsi@unimi.it; 14Department of Clinical and Experimental Pathology, Istituto Auxologico Italiano IRCCS, 20100 Milan, Italy

**Keywords:** abdominal obesity, obesity, BMI, BNT162b2 mRNA vaccine, antibody response, IgG-TrimericS, booster dose, COVID-19

## Abstract

Little is known about the long-term durability of the induced immune response in subjects with obesity, particularly in those with an abdominal distribution of adipose tissue. We evaluated SARS-CoV-2-specific antibody responses after BNT162b2 vaccine booster dose, comparing individuals with and without abdominal obesity (AO), discerning between individuals previously infected or not. IgG-TrimericS were measured in 511 subjects at baseline, on the 21st day after vaccine dose 1, and at 1, 3, 6, and 9 months from dose 2, and at 1 and 3 months following the booster dose. To detect SARS-CoV-2 infection, nucleocapsid antibodies were measured at baseline and at the end of the study. Multivariable linear regression evaluated the three-month difference in the absolute variation in IgG-TrimericS levels from booster dose, showing AO and SARS-CoV-2 infection status interactions (*p* = 0.016). Regardless of possible confounding factors and IgG-TrimericS levels at the booster dose, AO is associated with a higher absolute change in IgG-TrimericS in prior infected individuals (*p* = 0.0125). In the same regression model, no interaction is highlighted using BMI (*p* = 0.418). The robust response in the development of antibodies after booster dose, observed in people with AO and previous infection, may support the recommendations to administer a booster dose in this population group.

## 1. Introduction

People with chronic medical conditions have an increased risk of severe acute respiratory syndrome coronavirus 2 (SARS-CoV-2) infection and of developing major complications from coronavirus disease 2019 (COVID-19) [1]. Genetic and environmental host factors, including age, biological sex, comorbidities, and adipose tissue distribution converge to influence innate and adaptive immune responses to vaccines [2]. Individuals with obesity have an increased risk of contracting and developing a more severe case of COVID-19, especially those with a predominant accumulation of visceral adipose tissue (VAT), making these individuals an at-risk population [3,4,5]. Excessive VAT accumulation leads to low-grade systemic inflammation and predisposes individuals to an increased susceptibility to infection, increased morbidity and mortality, and reduced development of antibodies to vaccines due to a reduced immune response [6].

Moreover, abdominal adiposity is characterized by increased visceral and ectopic fat deposition, adipocyte dysfunction, inflammatory and adipokine dysregulation and insulin resistance as emerging risk factor for type 2 diabetes mellitus, hypertension, cardiovascular diseases, and fatty liver, conditions that could lead to more severe forms of infections, especially SARS-CoV-2 [5,7].

Approved messenger RNA (mRNA) vaccines against SARS-CoV-2 are highly effective at reducing infection and morbidity in the general population and have been highly recommended for individuals affected by obesity [8]. It is hypothesised that the chronic inflammatory state, immune dysregulation, and any related comorbidities, characteristic of subjects with obesity, particularly at the visceral level, predispose to a low immunological response to various vaccinations [5,9,10]. To date, the efficacy of vaccines does not seem to differ significantly in individuals affected by obesity or not [8,11]. Therefore, encouragement to receive vaccination is strongly advisable for people suffering from obesity [8]. However, the impact of obesity and abdominal obesity (AO) on the durability of mRNA vaccine-specific responses remains an open question [12,13].

We and others have previously reported a weaker immune response after two doses of BNT162b2 (Pfizer–BioNTech) and a greater drop in antibody levels at three months after dose 2 in infection-naïve subjects with AO compared with those without [14,15,16]. Therefore, the observed reduction in antibody levels over time after vaccination and the increase in positive cases has led to the need for additional booster vaccination [14,17,18,19,20].

The long-term duration of the immune response induced by the vaccination cycle and booster dose is still little understood, especially in patients with AO. In addition, uncertainties remain about the effect of vaccination on development of antibodies in individuals with previous infection [21,22,23,24,25].

To this end, we assessed SARS-CoV-2-specific antibody responses after the booster dose of the BNT162b2 mRNA vaccine in a cohort of health care workers. We compared the antibody response of individuals with or without AO, discerning their infection status (infection-naïve individuals and individuals infected pre or post vaccination cycle).

## 2. Material and Methods

### 2.1. Study Design and Population

In our study, the population was recruited in the prospective observational cohort study VARCO-19, that began in January 2021 and ended in March 2022 at IRCCS Policlinico San Donato, Italy. All participants gave written informed consent, and the protocol received approval from the IRCCS Lazzaro Spallanzani Ethics Committee (protocol code 48/2021/spall/PU/403-2021).

We collected blood samples from health care workers who received up to three doses of the BNT162b2 mRNA vaccine. The vaccination itself was not part of the study. A description of the enrolment process and inclusion criteria has previously been reported [14].

The study timeline is shown in Figure 1. 

### 2.2. Serological Testing

Serological testing was performed as previously described [14]. The Binding Antibody Unit (BAU)/mL was used as the unit of measurement for serological tests because it represents the unit of measurement imposed by the World Health Organisation (WHO) to standardise the values of antibodies that are measured with different methods in different laboratories around the world (≥33.8 BAU/mL is the minimum threshold of protection). The conversion factor from Arbitrary Unit AU/ML to BAU/mL is 2.6. Levels of antibodies were tested at eight time points: at baseline, 21 days following vaccine dose 1, and 1 (within 30–40 days), 3 (within 90–100 days), 6 (within 180–200 days), and 9 months (within 270–280 days) from vaccine dose 2. In addition, antibody levels were measured at 1 (within 30–40 days) and 3 months (within 90–100 days) from the booster dose. When the antibody determination was above the test range upper limit, a 1:20 dilution was performed using the specific buffer LIAISON^®^ TrimericS IgG Diluent Accessory (DiaSorin, Saluggia, Italy). Sporadic titres of >40,000 BAU/mL were obtained because of the limitation of the linear range at a dilution of 1:20. Since this was infrequent, no further dilutions were performed, and these samples were given a titre of 40,000 BAU/mL. At the beginning of the study, a qualitative evaluation of anti-nucleocapsid IgG (anti-N IgG) was performed to check whether SARS-CoV-2 infections had occurred prior to vaccination. To assess whether a participant had been infected with SARS-CoV-2 during the VARCO-19 study, we determined the anti-N IgG titre again three months after the booster dose.

### 2.3. Anthropometric Measures

Anthropometric measurements were evaluated at baseline. We determined waist circumference using a flexible tape measure by setting it midway between the iliac crest and the lower rib, to the nearest 1.0 cm. AO was defined as a waist circumference ≥ 102 cm in men and ≥88 cm in women [26]. Body weight was measured to the nearest 0.1 kg using a beam scale, and height was measured to the nearest 0.1 cm using a stadiometer. Body mass index (BMI) was calculated as the weight (km) divided by height (m) squared.

### 2.4. Statistical Analyses

Antibody levels were expressed as the geometric mean (±standard deviation, SD). Subjects were classified according to AO and BMI classes with or without a previous diagnosis of SARS-CoV-2 infection (no prior infection, infection diagnosis before vaccine or infection diagnosis after vaccine). Comparison of continuous values between groups was performed using the non-parametric Kruskal–Wallis test. Multivariable linear regression was used to account for possible confounding and to assess absolute variation: difference at three months of titre levels from booster dose in individuals with AO and without (or BMI-classes). The model was also adjusted for antibody levels from the booster dose, gender, age, smoking, hypertension, and previous diagnosis of SARS-CoV-2 infection and the interaction between previous infection and AO (or BMI classes). Least-squares (LS) means (±standard error, SE) were reported. The null hypothesis was rejected at *p* < 0.05. All statistical analyses were performed using SAS version 9.4 (SAS Institute, Cary, NC, USA).

## 3. Results

The initial study population consisted of 1060 employees of the IRCCS Policlinico San Donato, who received a BNT162b mRNA vaccine and from whom at least one blood sample was collected for antibody testing. The subjects enrolled in the VARCO-19 study were 41.4 ± 12.9 years old, 93% were Caucasian, and 62% were female: 1060 subjects who underwent vaccination (240 previously infected) provided samples 21 days following vaccine dose 1, after 1 month (within 30–40 days), and after 3 months (within 90–100 days) from dose 2; 977 (218 previously infected) provided samples at 6 months (within 180–200 days) after dose 2; 778 (177 with prior infection) provided samples at 9 months (within 270–280 days) after dose 2; 571 (126 with prior infection) provided samples at 1 month (within 30–40 days) after booster dose; 511 (109 prior infected individuals) provided samples after 3 months (within 90–100 days) following the booster dose (Figure 2).

In total, 511 individuals provided blood samples at all time points and were included in this analysis. Baseline characteristics of the patients are reported in Table 1.

We divided our sample according to two parameters: the presence of AO and the infection status. According to the waist circumference cut-off, 149 subjects (29.1%) were affected by AO and 362 (70.9%) exhibited normal adipose tissue distribution. According to infection status, subjects were divided into three categories: (1) those who had never been infected with SARS-CoV-2 (infection-naïve individuals, n = 262, 51.3%); (2) those who had developed the infection before the vaccine cycle (prior infected individuals, n = 109, 21.3%); and (3) those who developed the infection during the vaccine cycle (post infected individuals, n = 140, 27.4%). Figure 3 shows the IgG-TrimericS antibody response to mRNA SARS-CoV-2 vaccination in individuals with or without AO according to infection status at all time points.

### 3.1. Individuals Who Had Never Been Infected with SARS-CoV-2 (Infection-Naïve Individuals)

Among infection-naïve individuals, n = 262, 76 (29%) with and 186 (71%) without AO, between the third and ninth month after vaccine dose 2, there was a decrease in IgG-TrimericS levels in both subjects with AO and those without AO (0.22-fold [95% CI: 0.16–0.29] vs. 0.20-fold [95% CI: 0.17–0.23], respectively, Table 2, Figure 3). At one and three months after the vaccine booster dose, in subjects with AO, IgG-TrimericS levels were found to be lower than in individuals not suffering from AO without reaching statistical significance. An antibody peak was shown at one month after the vaccine booster dose (geometric mean BAU/mL ± standard deviation, 6470.55 ± 695.82 BAU/mL in individuals with AO vs. 7561.64 ± 406.80 BAU/mL in individuals without AO, *p* = 0.173), and a decline at the third month (2943.17 ± 335.19 BAU/mL in individuals with AO vs. 3346.92 ± 208.06 BAU/mL in individuals without AO, *p* = 0.413, Table 2, Figure 3).

### 3.2. Individuals Who Had Developed the Infection before the Vaccine Cycle (Prior Infected Individuals)

Among prior infected individuals, n = 109, 34 (31.2%) with and 75 (68.8%) without AO, between month three and month nine following the second dose of vaccine, in both subjects with AO and those without AO, the drop in IgG-TrimericS levels was significant (0.25-fold [95% CI: 0.15–0.42] vs. 0.20-fold [95% CI: 0.30–0.33], respectively, Table 2, Figure 3).

One month following the administration of the vaccine booster dose, a higher peak in IgG-TrimericS levels was reached in subjects affected by AO compared with those not affected by AO (6470.55 ± 695.82 BAU/mL vs. 4521.62 ± 429.76 BAU/mL, *p* = 0.0521). In the same two groups, between the first and third month after the booster dose, a similar drop in IgG-TrimericS levels was recorded (0.73-fold [95% CI: 0.48–1.14] vs. 0.74-fold, [95% CI: 0.57–0.96], *p* = 0.2352, Table 2, Figure 3).

### 3.3. Individuals Who Developed the Infection during the Vaccine Cycle

One hundred and forty individuals, 39 (27.9%) with and 101 (72.1%) without AO, became positive for SARS-CoV-2 anti-nucleocapsid IgG antibodies testing at the end of the observation period, indicating that they had contracted COVID-19 during the vaccine cycle. The most observed symptoms were mild, ranging from asymptomatic to mild fever. Among individuals who presented symptoms (n = 54), there were 38 (70.3%) individuals with AO and 16 (29.7%) individuals without AO. Between the third and ninth month after vaccine dose 2, there was a decrease in IgG-TrimericS levels in both subjects with AO and those without AO (0.27-fold [95% CI: 0.18–0.41] vs. 0.26-fold [95% CI: 0.21–0.3], respectively), similarly to infection-naïve individuals (Table 2, Figure 3). One month after the vaccine booster dose, individuals affected by AO exhibited comparable levels of IgG-TrimericS to those not affected by AO (6850.16 ± 947.21 BAU/mL vs. 5084.36 ± 500.43 BAU/mL, *p* = 0.119). Moreover, during the period from the first to the third month following the booster dose of the vaccine, individuals with AO achieved a higher peak of IgG-TrimericS levels compared with those not suffering from AO, even without statistical significance (8791.71 ± 1246.30 BAU/mL vs. 6944.18 ± 682.39 BAU/mL, *p* = 0.242, Table 2, Figure 3).

### 3.4. Multivariable Linear Regression Analysis

Multivariable linear regression, used to assess the difference at three months in the absolute change in IgG-TrimericS levels from the booster dose, showed evidence of an interaction between AO and SARS-CoV-2 infection (*p* = 0.016; Table 3). AO, in particular, was associated with a greater absolute variation in IgG-TrimericS in previously infected subjects regardless of sex, age, hypertension, or smoking, and IgG-TrimericS levels at the booster dose (LS means 8432.09 ± 1191.67 vs. 5091.93 ± 918.62, *p* = 0.0125). In the same regression model, no discernible interaction was identified when utilizing BMI classes (*p* = 0.418).

## 4. Discussion

In this longitudinal, observational study, VARCO-19, we presented data on antibody levels in a cohort of health care workers up to twelve months after a vaccination cycle with BNT162b2 mRNA and at month three following a booster dose.

In line with other studies, our findings showed a peak in antibody levels one month after the second vaccine dose, succeeded by a gradual decline until the administration of the booster dose. This trend was observed in individuals with or without AO, regardless of prior SARS-CoV-2 infection, whether before or during the vaccination cycle.

Previous studies have shown that humoral responses following dose 2 of the COVID-19 vaccine decrease in all population groups approximately six months after the second dose of a vaccine [18,27]. The decline in humoral responses does not depend on the type of COVID-19 vaccine administered, but on host factors [14,15,28,29,30,31]. Overall, it has already been shown that the humoral immune response to vaccination differs significantly between individuals previously infected with SARS-CoV-2 and naïve individuals [32].

We found that in all infection-naïve individuals, IgG-TrimericS concentrations decreased steadily after nine months from the second dose and after three months from the booster dose of a BNT162b2 m-RNA vaccine. Similarly to our previous study [14] and in according with other studies [16,33], infection-naïve individuals with AO, at one and three months after vaccine booster dose, have lower antibody levels than individuals without AO, suggesting that the duration of vaccine-induced immunity may be reduced in people with obesity. Watanabe et al. previously demonstrated an association between central obesity and a reduced adaptive response to the BNT162b2 m-RNA vaccine [29]. Furthermore, weight loss and/or improved metabolic health appears to reverse the effect. Adipose tissue, in addition to serving as a lipid store and energy source, is an endocrine organ that secretes fatty acids, metabolites, and adipokines, which have a crucial function in inflammation and the immune response, negatively influenced by the production of proinflammatory adipokines and cytokines [5,34,35,36,37]. In subjects affected by general obesity, and especially by AO, chronic inflammation, developed due to dysfunctional adipose tissue, negatively affects T-cell function, macrophage migration, and antibody response [38,39,40]. This is due to an overaccumulation of adipose tissue, particularly at the abdominal level, which can cause the secretion of adipokines and pro-inflammatory cytokines, which negatively impact the immune response [34].

Thus, immune dysfunction may increase the risk of developing SARS-CoV-2 infection and may decrease the response to the vaccine in naïve individuals with severe obesity [39,41].

We showed that, one month after vaccine booster dose, prior infected individuals with AO had higher concentrations of IgG-TrimericS than individuals without AO. This could be due to a more severe infection that is more frequent in patients with obesity [42,43].

Previous research, in fact, has shown that the level of antibodies to COVID-19 is associated with disease severity and that obesity correlates with a higher risk of contracting a more severe form of COVID-19 [44,45]. Subjects affected by severe obesity who survive COVID-19 generate robust and long-lasting SARS-CoV-2-specific T-cell immunity following severe infection; this is more evident in patients with obesity [46,47]. Furthermore, Muena et al. showed that immunization with CoronaVac or BNT162b2 vaccines, administered up to 13.3 months after the onset of COVID-19 symptoms, can significantly improve long-lasting neutralizing antibody responses induced by natural infection, suggesting that the infection induces a robust immune system response [48]. Hybrid immunity, resulting from the combination of prior SARS-CoV-2 infection and vaccination, appeared to confer enhanced protection against SARS-CoV-2 infections [49,50]. A booster dose administered after natural COVID-19 infection appears to provide a more consistent humoral immune response in terms of magnitude and quality than vaccination in infection-naïve individuals, fully consistent with clinical epidemiological observations [32]. This could explain the hyper-antibody response to the vaccine in this population, as reflected in our results.

Our data show that 27.4% of our population contracted COVID-19 despite having undergone the vaccination cycle and having high antibody levels. However, none of these subjects showed severe symptoms, but only mild manifestations such as fatigue, cold, sore throat, low-grade fever, and joint pain, suggesting that hybrid immunity could protect from severe COVID-19 illness. These results are consistent with a previous study showing that hybrid immunity and booster vaccination were associated with a lower risk of SARS-CoV-2 infection and fewer symptoms [51]. When ignoring the booster, any additional protection of two-dose vaccination in infection-naïve individuals was no longer observed [51]. Therefore, booster vaccination may reduce the risk of symptomatic SARS-CoV-2 infection in infection-naïve individuals, although this benefit appears to wane over time [51].

One of the limitations of our study includes the enrolment of only health care workers, who may not be representative of the general Italian population because they are much more exposed to the risk of SARS-CoV-2 infection. Another limitation of our study is the lack of measurement of virus-specific T cells. Anthropometric measurements were assessed only once. Furthermore, we did not evaluate proinflammatory markers of infection and do not know whether the vaccine BNT162b2 is protective against the Omicron XBB.15.

## 5. Conclusions

Abdominal obesity is a risk factor for severe COVID-19 complications. Currently, there are no reports of significant differences in COVID-19 vaccine effectiveness between people affected or not by obesity. Our results showed a robust response in the development of antibodies against COVID-19 after a BNT162b2 booster dose in people with AO who have been exposed to the virus. Our findings further support the recommendation that adults with high-risk medical conditions, including obesity, and, in particular AO, be vaccinated with booster doses, even if they have already come into contact with the virus.

Our results are based on a cohort of 511 patients; therefore, it may be interesting in future studies to evaluate antibody development after vaccine booster doses in a larger cohort more representative of the general population.

## Figures and Tables

**Figure 1 vaccines-11-01796-f001:**
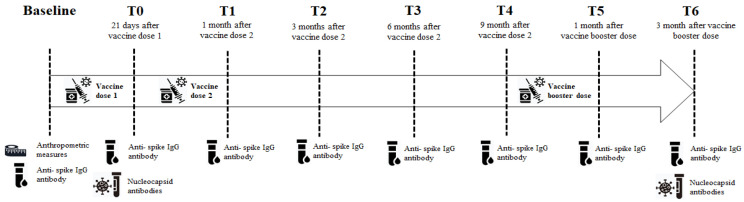
Study timeline. This figure shows all the events of the study.

**Figure 2 vaccines-11-01796-f002:**
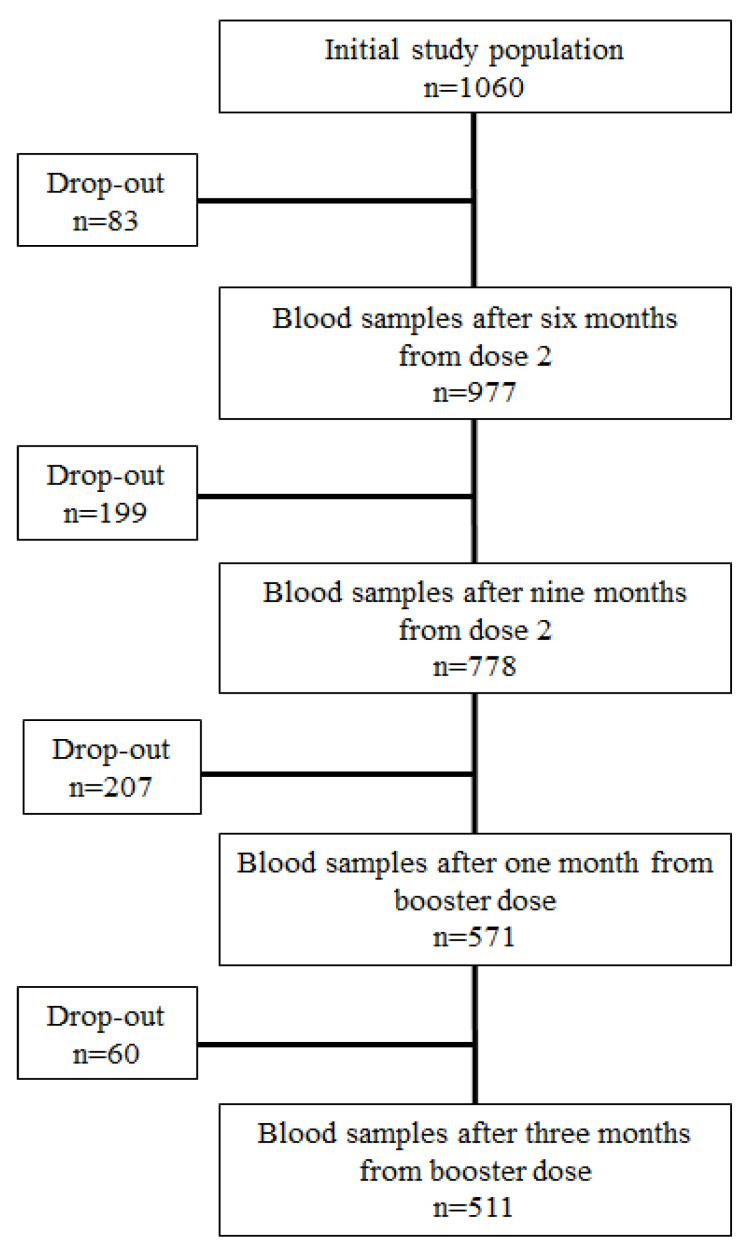
Study population flow diagram.

**Figure 3 vaccines-11-01796-f003:**
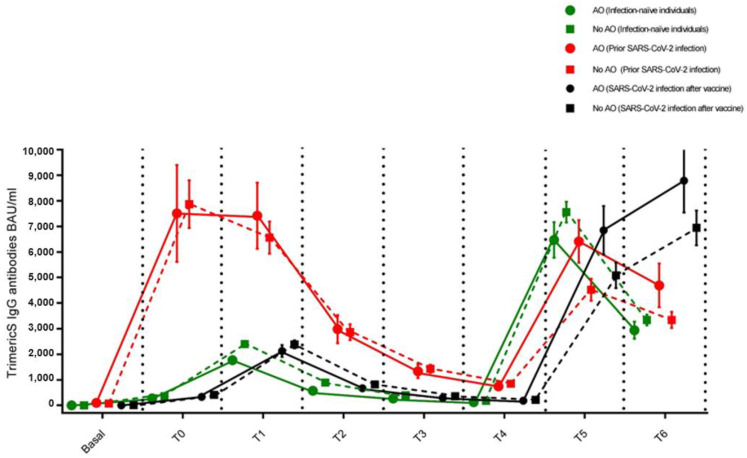
IgG-TrimericS antibody response to mRNA SARS-CoV-2 vaccination in subjects affected or not by abdominal obesity according to infection status. Infection-naïve individuals (green), individuals with prior infection (red), and individuals infected with SARS-CoV-2 after booster dose (black). Abdominal obesity (waist circumference ≥ 102 cm for men, ≥88 cm for women); no abdominal obesity (waist circumference < 102 cm for men, <88 cm for women).

**Table 1 vaccines-11-01796-t001:** Demographic and clinical traits at baseline of the entire study population stratified by the presence or absence of abdominal obesity and infection status.

		With Abdominal Obesity (n = 149)				Without Abdominal Obesity (n = 362)			
	Total (n = 511)	Without Prior SARS-CoV-2 Infection (n = 76)	With Prior SARS-CoV-2 Infection (n = 34)	SARS-CoV-2 Infection after Vaccine (n = 39)	*p*-Value	Without Prior SARS-CoV-2 Infection (n = 186)	With Prior SARS-CoV-2 Infection (n = 75)	SARS-CoV-2 Infection after Vaccine (n = 101)	*p*-Value
Age, years	44.03 ± 11.88	52.02 ± 10.64	47.26 ± 9.18	48.26 ± 8.28	0.0286	42.28 ± 12.02	41.73 ± 11.20	40.15 ± 11.70	0.3386
Ethnicity									
Caucasian	492 (96.28)	72 (94.74)	31 (91.18)	36 (92.31)		181 (97.31)	71 (94.67)	101 (100.00)	
Latin-American	13 (2.54)	3 (3.95)	3 (8.82)	3 (7.69)	0.7117 *	2 (1.08)	2 (2.67)	0 (0.00)	0.2074 *
African	2 (0.39)	0 (0.00)	0 (0.00)	0 (0.00)		2 (1.08)	0 (0.00)	0 (0.00)	
Arabic	4 (0.78)	1 (1.32)	0 (0.00)	0 (0.00)		1 (0.54)	2 (2.67)	0 (0.00)	
Gender									
Male	166 (32.49)	30 (39.47)	8 (23.53)	14 (35.90)	0.2656	56 (30.11)	28 (37.33)	30 (29.70)	0.4271
Female	345 (67.51)	46 (60.53)	26 (76.47)	25 (64.10)		130 (69.89)	47 (62.67)	71 (70.30)	
Smoking status									
Smoker	98 (19.18)	11 (14.47)	4 (11.76)	10 (25.64)	0.2128	36 (19.35)	11 (16.67)	26 (25.74)	0.1794
Non-smoker	413 (80.82)	65 (85.53)	30 (88.24)	29 (74.36)		150 (80.65)	64 (85.33)	75 (74.26)	
Comorbidities									
Hypertension	59 (11.55)	25 (32.89)	7 (20.59)	6 (15.38)	0.0944	12 (6.45)	3 (4.00)	6 (5.94)	0.7435
Diabetes mellitus	4 (0.78)	0 (0.00)	2 (5.88)	0 (0.00)	0.0509 *	0 (0.00)	1 (1.33)	1 (0.99)	0.2357 *
Cardiovascular diseases	17 (3.33)	6 (7.89)	0 (0.00)	1 (2.56)	0.2318 *	7 (3.76)	1 (1.33)	2 (1.98)	0.5268 *
Dyslipidaemia	32 (6.26)	9 (11.84)	4 (11.76)	5 (12.82)	1.000 *	7 (3.76)	3 (4.00)	4 (3.96)	1.000 *
Cancer	3 (0.59)	1 (1.32)	0 (0.00)	1 (2.56)	1.000 *	1 (0.54)	0 (0.00)	0 (0.00)	1.000 *
Anthropometric measurements									
Weight, kg	70.54 ± 15.12	84.08 ± 13.74	88.28 ± 16.96	80.86 ± 14.45	0.1026	64.38 ± 11.15	65.40 ± 12.29	65.53 ± 10.23	0.6481
Height, cm	167.63 ± 8.78	168.47 ± 10.54	166.14 ± 9.01	167.21 ± 8.61	0.4903	166.91 ± 8.39	168·21 ± 8.65	168.56 ± 8.10	0.2252
Waist, cm	85.78 ± 13.51	100.69 ± 8.76	103.26 ± 11.23	99.79 ± 9.47	0.2745	79.61 ± 9.20	76.63 ± 9.23	79.19 ± 9.22	0.9254
Waist male, cm	94.63 ± 11.85	107.25 ± 5.64	112.88 ± 9.79	107.93 ± 6.56	0.1104	89.26 ± 7.47	87.38 ± 7.93	87.75 ± 7.91	0.4994
Waist female, cm	81.52 ± 12.13	96.41 ± 7.74	100.31 ± 10.05	95.24 ± 7.64	0.0751	75.46 ± 6.31	75.02 ± 6.49	75.58 ± 7.13	0.8975
WHtR	0.51 ± 0.08	0.60 ± 0.05	0.62 ± 0.06	0.60 ± 0.05	0.0589	0.48 ± 0.05	0.47 ± 0.05	0.47 ± 0.05	0.5170
BMI, kg/m^2^	25.00 ± 4.45	29.53 ± 3.28	31.79 ± 4.39	28.81 ± 3.76	0.00019	23.02 ± 2.91	22.96 ± 2.87	23.02 ± 2.89	0.9869
BMI classes									
Underweight	20 (3.91)	0 (0.00)	0 (0.00)	0 (0.00)		12 (6.45)	2 (2.67)	6 (5.94)	
Normal weight	265 (51.86)	7 (9.21)	1 (2.94)	7 (17.95)	0.0951 *	129 (69.35)	51 (68.00)	70 (69.31)	0.8376 *
Overweight	156 (30.53)	34 (44.74)	13 (38.24)	20 (51.28)		43 (23.12)	22 (29.33)	24 (23.76)	
Obesity	70 (13.70)	35 (46.05)	20(58.82)	12 (30.77)		2 (1.08)	0 (0.00)	1 (0.99)	

* Fisher test. Abdominal obesity (waist circumference ≥102 cm for men, ≥88 cm for women); no abdominal obesity (waist circumference < 102 cm for men, <88 cm for women); waist-to-height ratio (WHtR); Body Mass Index (BMI); underweight (BMI < 18.5 kg/m^2^); normal weight (BMI from 18.5 to <25 kg/m^2^); overweight (BMI from 25 to <30 kg/m^2^); obesity (BMI ≥ 30 kg/m^2^). Antibody levels are expressed as BAU/mL (BAU, Binding Antibody Units) and are presented as the geometric mean [95% confidence interval]. Data are n (%), mean (SD).

**Table 2 vaccines-11-01796-t002:** IgG-TrimericS antibody levels of subjects who provided a blood sample at all time points stratified by the presence or absence of abdominal obesity and SARS-CoV-2 infection status.

		With Abdominal Obesity (n = 149)				Without Abdominal Obesity (n = 362)			
Antibody Levels	Total (n = 511)	Without Prior SARS-CoV-2 Infection (n = 76)	With SARS-CoV-2 Infection before Vaccine (n = 34)	SARS-CoV-2 Infection after Vaccine (n = 39)	*p*-Value	Without SARS-CoV-2 Infection (n = 186)	With SARS-CoV-2 Infection before Vaccine (n = 75)	SARS-CoV-2 Infection after Vaccine (n = 101)	*p*-Value
Baseline	9.67 ± 1.40 n = 90	4.81 n = 11	98.93 ± 74.44 n = 6	4.81 n = 8	<0.0001 *°	5.59 ± 0.58 n = 32	71.81 ± 24.26 n = 13	6.11 ± 1.19 n = 20	<0.0001 *°
21 days after dose 1	680.84 ± 49.58	274.16 ± 36.40	7511.83 ± 1895.77	333.58 ± 61.94	<0.0001 *°	362.54 ± 26.99	7872.56 ± 931.04	410.72 ± 32.39	<0.0001 *°
1 month after dose 2	2837.32 ± 110.30	1773.18 ± 155.93	7420.51 ± 1296.00	2130.24 ± 231.39	<0.0001 *°	2399.17 ± 123.48	6562.31 ± 631.09	2386.67 ± 156.08	<0.0001 *°
3 months after dose 2	1035.22 ± 43.80	575.90 ± 53.28	2985.81 ± 550.26	672.12 ± 68.08	<0.0001 *°	892.48 ± 44.54	2868.23 ± 308.43	820.87 ± 60.84	<0.0001 *°
6 months after dose 2	463.96 ± 21.32	267.78 ± 14.63	1335.15 ± 267.46	296.44 ± 30.03	<0.0001 *°	391.60 ± 22.71	1431.65 ± 155.39	358.84 ± 29.21	<0.0001 *°
9 months after dose 2	243.18 ± 11.93	126.78 ± 14.62	746.25 ± 131.40	183.41 ± 32.79	<0.0001 *°	176.38 ± 10.74	847.73 ± 92.92	216.88 ± 17.25	<0.0001 *°
1 month after booster dose	6218.25 ± 237.47	6470.55 ± 695.82	6413.54 ± 832.40	6850.16 ± 947.21	0.8892	7561.64 ± 406.80	4521.62 ± 429.76	5084.36 ± 500.43	<0.0001 #*
3 months after booster dose	4175.81 ± 181.19	2943.17 ± 335.19	4692.80 ± 854.60	8791.71 ± 1246.30	<0.0001 #°	3346.92 ± 208.06	3345.35 ± 314.00	6944.18 ± 682.39	<0.0001 #°

Abdominal obesity (waist circumference ≥ 102 cm for men, ≥88 cm for women); no abdominal obesity (waist circumference < 102 cm for men, <88 cm for women). * (SARS-CoV-2 infection before vaccine vs. no prior SARS-CoV-2 infection) # (SARS-CoV-2 infection after vaccine vs. no prior SARS-CoV-2 infection) ° (SARS-CoV-2 infection before vaccine vs. SARS-CoV-2 infection after vaccine).

**Table 3 vaccines-11-01796-t003:** Differences in absolute changes in IgG-TrimericS antibody levels from the booster dose up to three months assessed by univariate and multivariate linear regression (*p*-value).

	*p*-Value		
	Univariate	Multivariable According To Abdominal Obesity	Multivariable According to BMI Class
Sex	0.7120	0.5955	0.5656
Age	0.3000	0.4485	0.5412
IgG-TrimetricS antibody level at booster dose	0.0675	0.0855	0.0526
Prior SARS-CoV-2-infection	<0.0001	<0.0001	<0.0001
Abdominal obesity	0.1053	0.1092	-
*Interaction with prior SARS-CoV-2-infection * Abdominal obesity*		0.0163	-
BMI classes	0.0803	-	0.0821
*Interaction with prior SARS-CoV-2-infection * BMI classes*		-	0.4176
Smoking status	0.0925	0.3785	0.2968
Hypertension	0.0062	0.0020	0.0063
Diabetes mellitus	0.5878		
Cardiovascular diseases	0.7262		
Dyslipidemia	0.5087		
Cancer	0.3900		

Severe acute respiratory syndrome coronavirus 2 (SARS-CoV-2); abdominal obesity (waist circumference ≥ 102 cm for men, ≥88 cm for women); Body Mass Index (BMI).

## Data Availability

The data presented in this study are available upon request from the corresponding author.

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
