# Peer review of "BNT162b2 Booster Dose Elicits a Robust Antibody Response in Subjects with Abdominal Obesity and Previous SARS-CoV-2 Infection"

_vaccines, 2023, doi:10.3390/vaccines11121796_

Round 1

Reviewer 1 Report

Comments and Suggestions for Authors

This study is an interesting finding in the immunologic assessment of abdominal obesity. Moreover, the long-term follow-up in this study makes the result more useful and valuable than the other study that performed only short-term or mid-term follow-up.

This study is the extension outcomes in the long-term of the "Antibody responses to BNT162b2 mRNA vaccine: Infection-naïve individuals with abdominal obesity warrant attention" (Reference number 11). The methodology and assessment are well-designed and sound. The quality of this manuscript is acceptable for this journal and more interesting because this manuscript contained long-term follow-up outcomes.

Major concerns.

1. Is this study registered in any of the registries?
For example, the EU Clinical Trials Register (https://euclinicaltrials.eu), ClinicalTrials.gov or elsewhere.

I know this study was not an intervention with vaccine arms, but cohort or longitudinal studies can also register to the system. 

If it is, yes. Suggests adding the register with the identifier of this study/project to the 2.1.
If it is, no. Nothing to add a further statement.

2. Suggest adding the statement of the "conversion factor" of this instrument. This instrument reports as the AU/mL, not BAU/mL.
Not all readers are familiar with this instrument and may be confused about where BAU/mL comes from.

3. Suggest adding the cutoff of the serologic assessment tool to section 2.2.

Comments.

1. Introduction, the first paragraph.
You may add a piece more information that obese individuals may increase the risk of comorbidities, such as hypertension, diabetes or DLP, or may exist with these comorbidities.
The comorbidities related to obesity, such as type 2 Diabetes Mellitus, could substantially increase the severe conditions when getting the infection.
Adding this statement to emphasise why obesity must be vaccinated.

2. Line 63 "BNT162b2 mRNA vaccine Pfizer-BioNTech ". I think it is verbose.
Suggest using "BNT162b2 (Pfizer—BioNTech)" to make the vaccine and manufacturer names clear and distinct.

3. Have you tried to subgroup in the age group or every 10 years?
It is interesting to do it, and it can be proven that the elderly could have a lower immune response than the younger.
However, subgroups reduce the power of the test substantially. If it is significant, you must consider this point before adding it.

Author Response

REVIEWER #1

  1. This study registered in any of the registries? For example, the EU Clinical Trials Register (https://euclinicaltrials.eu), ClinicalTrials.gov or elsewhere. I know this study was not an intervention with vaccine arms, but cohort or longitudinal studies can also register to the system.  If it is, yes. Suggests adding the register with the identifier of this study/project to the 2.1.
    If it is, no. Nothing to add a further statement.

Thank you for this observation. Unfortunately, this study is not recorded in any of the registers mentioned. The protocol was approved by the IRCCS Lazzaro Spallanzani Ethics Committee (proto-col code 48/2021/spall/PU/403-2021).

  1. Suggest adding the statement of the "conversion factor" of this instrument. This instrument reports as the AU/mL, not BAU/mL. Not all readers are familiar with this instrument and may be confused about where BAU/mL comes from.

Thank you for this observation. The conversion factor from the AU/mL to BAU/ml is 2.6. If the results are expressed in AU/ml and you want to obtain the value in BAU/ml: multiply the value by 2.6. We add the conversion factor in the text from line 91 to line 96 as follows “The Binding Antibody Unit (BAU)/mL was used as the unit of measurement for serological tests, because it represents the unit of measurement imposed by the World Health Organisation (WHO) to standardise the values of antibodies that are measured with different methods in different laboratories around the world (≥33.8 BAU/ml is the minimum threshold of protection). The conversion factor from Arbitrary Unit AU/ML to BAU/mL is 2.6”.

  1. Suggest adding the cutoff of the serologic assessment tool to section 2.2.

Thank you for this suggestion. We have included the cutoff of the serological evaluation in line 95 as follows "≥33.8 BAU/ml is the minimum protection threshold".

  1. Introduction, the first paragraph. You may add a piece more information that obese individuals may increase the risk of comorbidities, such as hypertension, diabetes or DLP, or may exist with these comorbidities. The comorbidities related to obesity, such as type 2 Diabetes Mellitus, could substantially increase the severe conditions when getting the infection. Adding this statement to emphasise why obesity must be vaccinated.

Thank you for this observation. We enriched the text taking into account your suggestions from line 53 to line 70 as follows “People with chronic medical conditions have an increased risk of severe acute respiratory syndrome coronavirus 2 (SARS-CoV-2) infection and of developing major complications from coronavirus disease 2019 (COVID-19) [1]. Genetic and environmental host factors including age, biological sex, comorbidities and adipose tissue distribution converge to influence innate and adaptive immune responses to vaccines [2]. Individuals with obesity have an increased risk of contracting and developing a more severe case of COVID-19, especially those with a predominant accumulation of visceral adipose tissue (VAT), making these individuals an at-risk population [3–5]. Excessive VAT accumulation leads to low-grade systemic inflammation and predisposes to increased susceptibility to infection, increased morbidity and mortality and reduced development of antibodies to vaccines due to a reduced immune response [6]. Moreover, abdominal adiposity is characterized by increased visceral and ectopic fat deposition, adipocyte dysfunction, inflammatory and adipokine dysregulation and insulin resistance as emerging risk factor for type 2 diabetes mellitus, hypertension, cardio-vascular diseases and fatty liver, conditions that could lead to a more severe form of infections, especially SARS-CoV-2 [5,7].”.

  1. Line 63 "BNT162b2 mRNA vaccine Pfizer-BioNTech". I think it is verbose. 
    Suggest using "BNT162b2 (Pfizer—BioNTech)" to make the vaccine and manufacturer names clear and distinct.

Thanks for your observation. We have changed the sentence as suggested.

  1. Have you tried to subgroup in the age group or every 10 years? It is interesting to do it, and it can be proven that the elderly could have a lower immune response than the younger.
    However, subgroups reduce the power of the test substantially. If it is significant, you must consider this point before adding it.

We thank the reviewer for this very interesting suggestion. It is interesting to do it, and it can be proven that the elderly could have a lower immune response than the younger. However, subgroups reduce the power of the test substantially.

We tried to model the interaction between age, categorized by tertiles, and Prior SARS-CoV-2-infection in AO and not AO subjects. The interaction resulted not significant, namely p=0.2408 and p=0.0902 in AO and not AO, respectively. This result could be due to low power as, in fact, there are few young workers with AO.

Reviewer 2 Report

Comments and Suggestions for Authors

Investigators compare antibody response to BNT162b2 mRNA vaccine between individuals with abdominal obesity and individuals with normal body fat levels. They have observed that vaccinated individuals with abdominal obesity have higher antibody titer to the vaccine after natural infection in comparison to vaccinated individuals with normal body fat after natural infection. Overall, their manuscript is well written, data is clearly presented, and their methods are well described.  I have three main concerns with the study.  One, the study population is hospital employees in a large hospital in Northern Italy. This population is not representative of Italian population.  Due to the nature of COVID-19, hospital employees have been exposed to SARS-CoV-2 at a different level than normal population.  This should be stated in the limitations of the study. Second, main findings of the study are based on a small number of individuals that need to be substantiated in a larger cohort preferably not only in a non-random population of hospital employees. This should be clearly stated as well in the conclusions of the study. Third, I am a bit confused about the findings.  What would the investigators suggest for the role of higher Ab response in individuals with AO and natural infection? Is it protective or harmful?

I have few small comments to add.

Line 55: What do the authors mean by “poor immunological response”?

Line 73: I would not refer this cohort large.  I would just state what it is.

Line 79: Can you describe the cohort size here? Reader should not look for it from reference #11.

Line 114: AO esity - fix word

Line 118-119: absolute variation – difference?

Line 141: Title add “clinical characteristics at baseline

Line 141: Next page add description to the table if on a second page of the table

Author Response

Point-by-point reply

REVIEWER #2

  1. One, the study population is hospital employees in a large hospital in Northern Italy. This population is not representative of Italian population.  Due to the nature of COVID-19, hospital employees have been exposed to SARS-CoV-2 at a different level than normal population.  This should be stated in the limitations of the study.

Thanks for this observation. We modified the limitations of our work as follows “One of the limitations of our study includes the enrolment of only health care workers who may not be representative of the general Italian population, as they are much more exposed to the risk of SARS-CoV-2 infection”.

  1. Second, main findings of the study are based on a small number of individuals that need to be substantiated in a larger cohort preferably not only in a non-random population of hospital employees. This should be clearly stated as well in the conclusions of the study.

Thank you for this suggestion. We enriched the conclusions as follows “Since our results are based on a cohort of 511 patients, it may be interesting in future works to evaluate antibody development after vaccine booster doses in a larger cohort more representative of the general population”.

  1. Third, I am a bit confused about the findings.  What would the investigators suggest for the role of higher Ab response in individuals with AO and natural infection? Is it protective or harmful?

Individuals with abdominal obesity and previous infection, after having undergone the complete vaccination cycle and the booster dose, develop a hybrid immunity which appears to provide a more consistent humoral immune response in terms of entity probably due to a more serious infection than it is more frequent in subjects affected by obesity.

  1. Line 55: What do the authors mean by “poor immunological response”?

A “poor immunological response” refers to an inadequate or weakened reaction by the immune system in antibody development following the BNT162b2 mRNA vaccine. We modified the sentence as follows “It is hypothesized that the chronic inflammatory state, immune dysregulation and any related comorbidities, characteristic of subjects with obesity, particularly at the visceral level, predispose to a low immunological response to various vaccinations”.

  1. Line 73: I would not refer this cohort large. I would just state what it is.

We agree with your suggestion, we deleted the adjective “large” referring to the cohort.

  1. Line 79: Can you describe the cohort size here? Reader should not look for it from reference #11.

Thanks for this observation. We described the cohort size in the results of the work.

  1. Line 114: AO esity - fix word

Thanks for letting us know about this typo. We fixed it.

  1. Line 118-119: absolute variation – difference?

Thanks for this observation. We modified the sentence as follows “We used multivariable linear regression to account for possible confounding and to evaluate the absolute variation: three-months difference of titre levels starting from booster dose in individuals with and without AO (or BMI-classes).”

  1. Line 141: Title add “clinical characteristics at baseline

Thanks for this observation. We modified the table title as follows “Demographic and clinical characteristics at baseline of all study population with or without abdominal obesity according to infection status”.

  1. Line 141: Next page add description to the table if on a second page of the table

We left the table description at the end of the table on the second page.
